# Incorporating Immunotherapy in the Management of Gastric Cancer: Molecular and Clinical Implications

**DOI:** 10.3390/cancers14184378

**Published:** 2022-09-08

**Authors:** Alessandro Agnarelli, Viviana Vella, Mark Samuels, Panagiotis Papanastasopoulos, Georgios Giamas

**Affiliations:** Department of Biochemistry and Biomedicine, School of Life Sciences, University of Sussex, Falmer, Brighton BN1 9QG, UK

**Keywords:** gastric cancer, immunotherapy, PD-L1, *H. pylori*, EBV, immune microenvironment

## Abstract

**Simple Summary:**

Gastric cancer is one of the most common malignant tumours worldwide, with the fifth and third highest morbidity and mortality, respectively, of all cancers. Survival is limited, as most of the patients are diagnosed at an advanced stage, and are not suitable for surgery with a curative intent. Chemotherapy has only modestly improved patients’ outcomes and is mainly given with a palliative intent. Immunotherapy has improved overall survival of patients with gastric cancer, and has thus become a new standard of care in clinic. In this review we discuss the strong molecular rationale for the administration of immunotherapy in this disease and analyse the clinical data supporting its use.

**Abstract:**

Gastric cancer has a median survival of 11 months, and this poor prognosis has not improved over the last 30 years. Recent pre-clinical data suggest that there is high tumour-related neoantigen expression in gastric cancer cells, suggesting that a clinical strategy that enhances the host’s immune system against cancer cells may be a successful approach to improve clinical outcomes. Additionally, there has been an increasing amount of translational evidence highlighting the relevance of PD-L1 expression in gastric cancer cells, indicating that PD-1/PD-L1 inhibitors may be useful. Several molecular subgroups of gastric cancer have been identified to respond with excellent outcomes to immunotherapy, including microsatellite instable tumours, tumours bearing a high tumour mutational burden, and tumours related to a chronic EBV infection. In gastric cancer, immunotherapy has produced durable responses in chemo-refractory patients; however, most recently there has been a lot of enthusiasm as several large-scale clinical trials highlight the improved survival noted from the incorporation of immunotherapy in the first line setting for advanced gastric cancer. Our review aims to discuss current pre-clinical and clinical data supporting the innovative role of immunotherapy in gastric cancer.

## 1. Introduction

Gastric cancer (GC) is the fifth most commonly diagnosed cancer worldwide and the fourth most common cause of cancer death [1]. As GC is often diagnosed at an advanced stage, mortality remains very high. GC shows both genetic and environmental risk factors, with Helicobacter pylori (*H. pylori*) infection being the most well-described risk factor leading to GC. Germline genetic alterations are also involved in 1–3% of cases [2]. The 5-year survival rate of GC is only around 20–30%. This is largely due to the late stage at which the disease is normally diagnosed as pre-metastatic GC carries a more favourable survival rate around 67% [3].

Due to the poor survival rates of GC, immunotherapy has been widely explored as a potential treatment [4]. Both active and passive immunotherapies have been examined. Active immunotherapies involve using a patient’s own immune system to treat the disease whereas passive immunotherapies rely on exogenous agents administered to patients such as antibodies in order to treat the tumour [4]. The great efficacy observed in melanoma has propelled immunotherapies to be explored in a variety of other tumours, particularly breast, prostate, and lung cancer [5]. This review looks at how immunotherapy has been applied to GC, particularly focusing on immune checkpoint inhibitors, which have become part of the standard of care management recently.

## 2. Molecular Classification

The earliest classification system of GC is the Lauren classification which separates gastric cancer into intestinal and diffuse subtypes [6]. Intestinal tumours are more adhesive than diffuse, forming tubular or glandular formations. They are more common in older patients and associated with a more favourable prognosis. Intestinal adenocarcinoma is more often associated with *H. pylori* infection. Diffuse GC cells are less adhesive and typically occur in younger patients [7]. Diffuse type GC is associated with a poorer prognosis than intestinal type [8].

In 2020, the WHO classified GC into four main histological subtypes: tubular, papillary, mucinous, and poorly cohesive (including signet ring cell carcinoma) [9]. Tubular adenocarcinoma is the most common subtype of early GC and is characterised by irregular tubules, frequently found with intraluminal mucous [10]. Papillary cancers more often affect older patients and present as epithelial projections on a fibrovascular core. Mucinous adenocarcinomas are composed of extracellular mucinous pools, comprising greater than 50% of the tumour bulk. Poorly cohesive carcinomas are formed of signet ring cells and other ring cells, often resulting in lymph node invasion [9]. In addition to these subtypes, other less common variants exist which do not fit into the above categories.

A study in 2014 by the Cancer Genome Atlas Research Network classified gastric adenocarcinoma (GAC) into four new categories: Epstein–Barr virus (EBV)-positive, Microsatellite instability (MSI), Genomically stable (GS), and chromosomal instability (CIN) [11] (Figure 1). These novel categories provide a potential method for the stratification of patients to enable the use of targeted therapies.

EBV-positive cancers comprise around 9% of cases and are characterised by increased levels of DNA hypermethylation, including *CDKN2A* promoter hypermethylation [12]. *PIK3CA* mutations are another common feature of EBV-positive GC [12]. Other molecular features of EBV-positive GC are *ARID1A* and *BCOR* mutations, whereas the mutation rate in *TP53* is very low. EBV-positive cases are generally diagnosed in males. PD-L1/L2 overexpression is also seen [13], indicating a role in interacting with immune cells.

MSI cancers are diagnosed in more elderly patients and slightly more frequent in females [14]. They display higher rates of mutation and hypermethylation, including at the *MLH1* promoter. MSI tumours form 22% of those considered in this study. Other mutations include *PIK3CA*, *ERBB3*, *ERBB2*, *EGFR,* and *MHCI*-related genes such as *B2M* and *HLA-B*. It was suggested that these MHCI mutations may provide MSI tumours with an advantage by reducing immunogenicity [11].

GS cancers are comprised mainly of diffuse type tumours based on the Lauren classification. They present at an earlier age than other cancers and are enriched in *CDH1*, *RHOA,* and *ARID1A* mutations. *CLDN18*-*ARHGAP* fusions are also seen in genomically stable cancers [15]. The authors noted that despite the mutations previously discussed, there were few clear treatment strategies that could be explored for these patients [11].

Fifty percent of the cancers studied were classified as CIN which presented at a higher rate in the gastroesophageal junction/cardia compared to the other subtypes. *TP53* mutations were common and found in 71% of the tumours. Additionally, *EGFR* amplification was observed as well as an increase in EGFR Y1068 phosphorylation [11]. The authors suggested that the frequent RTK alterations seen in CIN tumours may enable treatments focusing on VEGF and EGFR inhibition to be explored. CDK inhibitors may also be useful given the high rate of cell cycle protein (*CCND1*, *CCNE1 CDK6*) amplification [11].

## 3. The Role of Tumour-Infiltrating Immune Cells in the Gastric Cancer Microenvironment

Tumour-infiltrating immune cells are important components of the tumour microenvironment with many described roles. Numerous studies have reported interactions between solid tumours and their immune microenvironment promoting invasion and metastasis. In GC, the main immune cell types in the microenvironment are tumour-associated macrophages (TAMs) and tumour-infiltrating lymphocytes (TILs).

TAMs are an important immune cell subtype in GC. Derived from lymphatic and blood monocytes, TAMs infiltrate the tumour and secrete various chemokines to regulate cell growth, invasion, and metastasis [16]. TAMs are typically described as having an M1 or M2 polarisation, with classical M1 TAMs being anti-tumourigenic and pro-inflammatory, producing IL-1β, IL-1α, IL-12, and TNF-α, whereas alternative M2 TAMs display anti-inflammatory activity and immunoregulation, promoting tumourigenic functions through IL-4, IL-6, and IL-10 [17].

Various properties of TAMs have been proposed as prognostic biomarkers in GC [17]. These include TAM density [18] and M2 infiltration [19] which are associated with poor prognosis and M1 infiltration which correlates with better prognosis [19]. Various secreted factors associated with TAMs are also potential biomarkers, such as Tim-3 which correlates with increased tumour invasion and lymph node metastasis [20] and CCL5/RANTES and NFKB1 where SNPs are associated with altered clinical outcome [21].

Importantly, TAMs induce immune tolerance in GC where programmed cell death protein 1 (PD-1) and cytotoxic T-lymphocyte antigen 4 (CTLA-4) promote an immunosuppressive microenvironment by blocking cytotoxic T cell anti-tumour activity. Macrophages can induce PD-L1 expression in GC cells through TNF-α and IL-6 signalling [22]. These pro-inflammatory cytokines can regulate STAT3 and NF-kB signalling in tumour cells, inducing PD-L1 expression and contributing to immunosuppression in tumours. TAMs are also known to play important roles in angiogenesis where they accumulate in hypoxic regions in tumours [23,24]. TAM infiltration correlates highly with PD-L1 expression, impacting metastasis and survival rate [25].

Natural killer (NK) cells are important players in GC as they can often attack cancer cells after the tumours have escaped detection by CD8+ T cells [26]. NK cells work by releasing granules containing perforin and granzymes to cause cancer cell lysis. They also express TNF-related apoptosis-inducting ligand family (TRAIL) and Fas ligand (FASL) inducing apoptosis in cancer cells [27]. Cytokine production (IFN-γ and TNF-α) also increases the cytotoxic anti-tumour response. In GC, NK cells are also able to target CD133+ cancer stem cells [28]. Despite this, as GC progresses, NK activity decreases through increased apoptosis [29], upregulation of inhibitory receptors and downregulation of activating receptors [30], a decrease in cytotoxic granule production and cytokine release [31], and a reduction in infiltration [32].

Dendritic cells (DCs) are another important cell type in GC. They present cancer antigens to immune cells to regulate the immune response against tumour cells. DC infiltration has been associated with an increased 5-year survival rate in GC [33]; however, CD83+ DCs are associated with a poorer prognosis in GC, both in the primary tumour and in lymph nodes [34]. Clinical studies have explored the use of DCs pulsed with tumour-associated antigens, followed by autologous transplant into patients, for instance HER2 peptide-activated DCs could induce a T cell response against the antigen [35]. Additional DC therapies in GC are reviewed in Tewari et al. [36].

Another key population of immune cells in GC is TILs. These cells have prognostic significance in gastric cancer. TILs are comprised of T cells, B cells and NK cells [37]. The anti-tumour immune response occurs when tumour-specific antigens are processed by DCs and presented to T cells. Stromal TILs have previously been found as predictors of poorer disease-free and recurrence-free survival in GC [38] whereas intratumoural TILs are associated with increased overall survival and cancer-specific survival in EBV-associated GC [39,40].

Different lymphocyte subpopulations have been studied in GC, particularly in terms of prognostic relevance. For instance, a high density of CD8+ lymphocytes was found to increase overall survival (OS) in GC [41], whereas high numbers of Th22 and Th17 cells was associated with a decrease in OS [42]. These are summarised in Kang et al. [37].

Immune evasion is an important step in cancer progression, where cancer cells avoid detection by expression of PD-L1. T lymphocytes detect cancer cells through interacting with MHC on the tumour cell through the T-cell receptor. PD-L1 expression on the tumour cells, however, can bind to PD-1 on lymphocytes, inhibiting the anti-tumour immune response [43]. In GC, around 30% of patients have tumour cells positive for PD-L1 and 50% have PD-1 expression, largely on TILs [44]. There is evidence to suggest a potentially prognostic role for the expression of PD-L1 in GC, indicating that immune checkpoints are an important pathway in GC progression [44]. As such, they may be useful targets in the treatment of GC, as discussed later in the review.

## 4. Role of Chronic Infection with *Helicobacter pylori* and the Association with an Immunosuppressive Microenvironment

Helicobacter pylori (*H. pylori*) is a facultative, spiral-shaped, Gram-negative bacterium that selectively colonises the gastrointestinal mucosa [45,46,47]. Four decades after its discovery in 1982, *H. pylori* represents a well-established risk factor for GC, and it is recognised as a type I carcinogen by the International Agency for Research on Cancer (IARC), with 90% of non-cardia gastric cases attributable to this bacterium [47,48,49,50].

About half of the world’s population is infected with *H. pylori*, and its prevalence is widely variable across different geographical regions [51]. An increased prevalence has been reported in less developed countries as compared to industrialised regions where a progressive decline of *H. pylori* infection has been registered in recent decades, also resulting in a decreased GC incidence [50].

In most cases, individuals infected with *H. pylori* are asymptomatic. However, the majority of them concomitantly develop chronic inflammation and have a higher risk of several upper gastrointestinal diseases, such as peptic ulcer disease, gastric atrophy, gastric adenocarcinoma, and primary gastric lymphoma [46,52].

Despite the apparent association with *H. pylori* infection, the risk of developing such malignancies is dependent on the interactions occurring between the pathogen and its host: while *H. pylori*-mediated infection can trigger a number of effects in the host, including induction of inflammatory response and ultimately lead to neoplastic transformation, the growth and ability of certain bacterial strains to colonise the host can, in turn, be influenced by the host immune response [53]. In fact, the *H. pylori* population can be exceptionally diverse with distinct, often coexisting, bacterial strains within the gastric microbiota. The host genotype can provide a selective pressure for *H. pylori* strains having better fitness under hostile habitat conditions, therefore increasing the risk of carcinogenesis. In this dynamic scenario, alterations to the ‘host–bacterial equilibrium’ by a number of factors such as specific bacterial strain, host inflammatory response, and particular pathogen–host interactions can differently affect the risk of developing gastric tumours [45,53]. This also explains why only some of the carriers eventually develop these malignancies.

### 4.1. Contribution of H. pylori Virulence Factors to Chronic Inflammation

During *H. pylori* infection, stimulation of several inflammatory signals is key to the establishment of an inflammatory environment in the gastric epithelium. This marks an important step towards initiation of a more complex inflammatory and immune response which can eventually culminate in the development of peptic ulceration and gastric malignancies.

A range of genes expressed by *H. pylori* are involved in the infection and remodelling process of the microenvironment [45,54,55]. These include ureases that confer this pathogen the ability to colonise and neutralise the highly acidic environment found within the stomach by converting urea into ammonia, therefore establishing the optimal pH conditions for its growth [54]. The subsequent increase in pH contributes to alter the viscosity of gastric mucus facilitating *H. pylori* diffusion through the mucosal barrier and allowing the pathogen to gain access to the underlying epithelial cells [56]. This event is crucial for the gastric epithelium colonisation process, which is the basis of the inflammatory reaction induced by *H. pylori*. Moreover, urease could contribute to gastric carcinogenesis by producing reactive oxygen species and activating the lipoxygenase pathway, resulting in differentiation of endothelial cells [57,58].

*H. pylori* flagella favour the colonisation of the gastrointestinal mucosa and contribute to bacterial motility [59]. FlaA is one of the main structural proteins of the *H. pylori* flagellum which can evade the host immune response, as it can escape recognition by the Toll-like receptor 5 (TLR5), a member of the Toll-like receptor family that normally recognises most bacterial flagellins [59,60]. As flagellins are critical for persistent *H. pylori* colonisation, and given that *H. pylori* colonisation is the basis of inflammation, flagella can be considered responsible for both inflammation and immune evasion [46,59].

Other players can also act in a cascade of events inducing damage to the gastric mucosa and host inflammatory response. Amongst these, there is the cytotoxin-associated gene CagA which is part of the cag pathogenicity island (cag PAI)—locus of approximately 40 kb containing 31 genes, the majority of which encode for the cag secretion system (T4SS) [46,60]. Importantly, cag Pal+ *H. pylori* strains increase the risk of gastritis, atrophic gastritis, and GC than strains lacking the cag island [61]. CagA modulates the host cell signalling both in a phosphorylation-dependent and independent manner. Phosphorylation of CagA has been reported to induce sustained activation of the ERK1/2 MAP kinase and NF-kB signalling pathways, and disruption of epithelial cell tight junctions with damage to the gastric mucosa [57]. Activation of proinflammatory responses, mostly through IL-8 pro-inflammatory cytokine production instead, appears to be independent of CagA phosphorylation [46,51,60].

Besides cagPAI-induced IL-8 secretion, expression of the *H. pylori* outer inflammatory protein (OipA) has also been shown to be involved in IL-8 production, supporting its importance in the inflammation process [60]. Similarly, the duodenal ulcer promoting gene A (dupA) was also shown to be significantly associated with the secretion of IL-8 [62]. The blood group antigen binding adhesin (BabA2) which is amongst the most well described *H. pylori* outer membrane proteins (OMPs) is associated with duodenal ulcer disease and GC. It binds to the antigen Lewis-expressing gastric epithelial cells and it is essential to maintain persistent *H. pylori* colonisation [63]. Other outer membrane proteins include AlpA, AlpB and HopZ, and SabA, the latter being responsible for binding to sialylated receptors on neutrophils and inducing neutrophil activation [47].

Other bacterial virulence factors are the vacuolating cytotoxins (VacA), a secreted protein that penetrates the epithelial cell membrane, induces vacuole formation and apoptosis through mitochondrial damage and induces inflammatory changes to the gastric mucosa [47,51]. Although VacA has numerous effects on epithelial cells, these induce cellular impairment rather than pro-inflammatory cytokine release.

### 4.2. Host Immune Response to H. pylori Infection

During *H. pylori* infection, both innate and acquired immune responses are intensely stimulated. Despite the strong immune responses, *H. pylori* has the remarkable ability to persist for a very long time in the gastric mucosa, actively modulating and evading the host response to establish an immunosuppressive environment that maintains chronic infection [52].

Most of the factors mentioned above are thought to intensify local inflammation with consequent infiltration of inflammatory cells to the gastric mucosa. Innate host defence mechanisms are triggered as a first line of defence and are crucial to increase risk of gastric carcinogenesis and severity of the disease [60]. The innate immune response includes the nucleotide-binding oligomerisation domain protein 1 (Nod1) [64]. This is a pattern recognition receptor (PRR) responsible for the recognition of *H. pylori* peptidoglycan components secreted by the cag secretion system, and activation of NF-kB-dependent proinflammatory responses. The most studied PRRs are the Toll-like receptors (TLRs), expressed on epithelial and innate immune cells, which interact with diverse *H. pylori* antigens (including lipoteichoic acid, lipoproteins, lipopolysaccharide, and flagellin) and initiate the adaptive immune responses [65]. The ability of *H. pylori* to escape TLRs recognition is well described, as well as for the RIG-I like receptors (RLRs) and C-type lectin receptors (CLRs), such mechanisms are concisely reviewed by Karkhah et al. [52,66].

Amongst the main players of the innate response to *H. pylori*, there are macrophages which, along with monocytes and DCs, are responsible for the recruitment of lymphocytes and the stimulation of T-helper (Th) cell-specific responses by releasing factors such as IL-12. In particular, stimulation of Th1 cells is predominant, and it is central in the T-cell response to *H. pylori*, as it produces cytokines such as IFN-γ and leads to pro-inflammatory cytokine release, for example TNFα, and interleukins [67] (Figure 2).

Upon intense stimulation, CD4+ helper T cells and CD8+ killer T cells are recruited to the gastric mucosa, with activation of CD4+ T cells [65]. In this scenario, other CD4+ T cell subsets, such as Th17 and Tregs also play a role: while Th1 and Th17 cells enhance gastric inflammation, Tregs seem to exhibit a protective role against inflammation and contribute to bacterial persistence [52]. In fact, regulatory T cells are responsible for the secretion of suppressive cytokines (IL-10 and TGFβ), through which, they negatively modulate immune and inflammatory responses, facilitating persistent bacterial colonisation [60].

Immunosuppression by *H. pylori* is elicited through several mechanisms, including direct inhibition of T-cells. In particular, VacA interferes with T-cell responses by preventing MHC class II-mediated antigen presentation and T-cell activation. It also inhibits T cell proliferation and promotes mitochondrial apoptosis [68]. *H. pylori* can also evade the bactericidal activity of macrophages and survive inside larger than normal phagosomes, referred to as megasomes [46,69].

In summary, failure to maintain a balance between the persistent activation of inflammatory responses on the one hand, and the ability of *H. pylori* to shield itself and subvert host immunity on the other hand, is crucial for the progression to disease. *H. pylori* chronic infection results in a profound alteration of the gastric microenvironment where inflammation and mucosal damage persist over time, eventually prompting the surrounding normal cells to undergo malignant transformation.

### 4.3. H. pylori Infection Promotes an Immunosuppressive Tumour Microenvironment

As discussed above, the establishment of *H. pylori*-mediated chronic infection has been shown to have profound implications on the host immune system and its functions. In fact, by exerting immunomodulatory effects on a variety of immune cells (through negative modulation of Th1 and Th17 cells, macrophages and dendritic cells, and promotion of Treg cells activity) *H. pylori* is thought to be responsible for the creation of an immunosuppressive environment [70,71].

Under normal conditions, the immune system retains the ability to generate tumour-specific immune responses to thwart cancer formation and progression, a concept known as anti-tumour immunity [72]. Such anti-tumour immunity can be boosted by cancer immunotherapies, for instance by using immune checkpoint inhibitors (ICIs) that block the natural function of immune checkpoints and prevent attenuation of immune cell activation [71]. These include inhibitors of the cytotoxic T-lymphocyte antigen 4 (CTLA-4), or the programmed cell death protein 1 (PD-1)/programmed cell death ligand 1 (PD-L1) which will be extensively described later in this review [73].

Moreover, metabolic reprogramming, arising from *H. pylori* infection has been shown to drive immunosuppression [74]. Infection with *H. pylori* upregulates Lon protease 1 expression and increases glycolysis [75]. One study found that in mice gastric macrophages, *H. pylori* increases arginase II (Arg2) production and reduces NO production, thereby limiting the bactericidal and inflammatory response of macrophages [76]. Another interesting mechanism occurs when the gastric pH rises in chronic atrophic gastritis, enabling new bacteria to colonise the stomach [77]. Some of these bacteria, including Stenotrophomonas and Selenomonas correlated with the presence of the immunosuppressive cells, BDCA2+ plasmacytoid dendritic cells and Foxp3+ Treg cells [78]. The presence of these immunosuppressive cells can promote immune evasion in GC, contributing to disease progression.

Accumulating evidence has revealed that the diverse composition of the gut microbiota can impact the response of subgroups of patients to cancer immunotherapy [70,79,80]. While different bacterial species of the gut microbiota have been shown to improve the effectiveness of immune checkpoint inhibitors [81] a negative impact of *H. pylori* infection on the efficacy of cancer immunotherapies has been described in patients with non-small-cell lung cancer (NSCLC), where *H. pylori* seropositivity was associated with poor response to anti-PD-1 immunotherapy [71]. Moreover, Oster P et al. also found that H. pylori infection reduces the cancer immunotherapy efficacy of anti-CTLA4 and/or anti-PD-L1 therapy in the MC38 colon adenocarcinoma pre-clinical model. Mechanistically this was explained by a defective dendritic cell’s activation induced by *H. pylori*, and the resulting attenuation of the anti-tumoural CD8+ T cell responses [71]. A summary of the studies reporting the role of PD-L1 in *H. pylori* infection reviewed by Silva et al. suggests an increased PD-L1 expression following *H. pylori* infection. Up-regulation of PD-L1 is also thought to result in an inadequate immune surveillance, favouring tumour escape and progression [82,83].

As argued by Shi et al., the *H. pylori* negative modulation of the immune system and up-regulation of PD-L1 suggest that *H. pylori* may affect the efficacy of immunotherapy with PD-1/PD-L1 inhibitors in GC. Nevertheless, a possible correlation between *H. pylori* and the effectiveness of PD-1/PD-L1 inhibitors has only been addressed in a limited number of studies, and to date there is no conclusive evidence on whether *H. pylori* can be considered as a predictive biomarker for response to immunotherapy in GC.

## 5. Molecular Mechanisms Underlying the Pathology of Microsatellite Instable Gastric Cancer

Genomic instability is one of the major hallmarks of cancer development [84]. It is believed to be one of the initial steps of gastric carcinogenesis and can be found in all different histological subtypes of GC [85]. Microsatellites (MS) are repetitive and specific DNA sequences characterized by a high mutation rate [86,87]. Microsatellite instability (MSI) is a hyper-mutable phenotype caused by a non-functional DNA mismatch repair (MMR) machinery at MS sites. During DNA replication, the insertion or deletions of nucleotides in MS regions because of germline mutations or epigenetic silencing cause malfunction of MMR system [88,89].

The MMR system includes several proteins: human MutL homolog 1 (hMLH1), human MutL homolog 3 (hMLH3), human MutS homolog 2 (hMSH2), human MutS homolog 3 (hMSH3), human MutS homolog 6 (hMSH6), human post meiotic segregation increased 1 (hPMS1), and human post meiotic segregation increased 2 (hPMS2) [87]. During DNA replication, hMSH2/hMSH6 and hMSH2/hMSH3 complexes are responsible for detecting and binding small DNA mismatch errors while the excision and re-synthesis of the corrected DNA bases in the mismatch site is detected by the heterodimeric complex hMLH1/hPMS2. Defects in one or more MMR machinery elements determine the unsuccessful repair of the DNA [87]. Different processes including promoter methylation, chromosomic rearrangements that lead to loss of heterozygosity or mutations in the coding region are responsible for the inactivation of MMR proteins [90,91]. The main cause of MMR deficiency in both sporadic and familial MSI GCs is represented by the hypermethylation of *hMLH1* promoter [92,93]. Conversely, mutations of *hMLH1* and *hMSH2* are relatively rare (15% and 12%, respectively) [94]. Some reports showed that MSI represents an early molecular event during gastric carcinogenesis [95,96]. However, Ling et al. reported that the promoter methylation of *hMLH1* represents a later event during the natural process of tumour growth and the time-dependent acquisition of MSI may be due to the *hMLH1* silencing [97]. Both sporadic GC and Lynch syndrome (LS) show MSI [91,98]. LS is mainly caused by autosomal dominant mutations affecting *hMLH1* and *hMSH2* and less frequently *hPMS2* and *hMSH6* [91]. Moreover, the epigenetic silencing of *hMSH2* by a constitutional 3′ end deletion of *EPCAM* can also cause LS [99,100]. Patients affected by LS show an increased predisposition to develop GC at a younger age (11.3-fold in the 30s and 5.5-fold in the 40s) [90,91].

High microsatellite instability (MSI-H) GC has been reported to be associated with mutations in genes involved in different cellular processes like cell cycle regulation and apoptosis (e.g., *TGFβ RII*, *IGFIIR*, *TCF4*, *RIZ*, *BAX*, *CASPASE5*, *FAS*, *BCL10*, and *APAF1*) or genomic integrity maintenance (e.g., *MSH6*, *MSH3*, *MED1*, *RAD50*, *BLM*, *ATR*, and *MRE11*) [89]. Furthermore, some studies reported the important role of the phosphoinositide3-kinase (PI3K)/AKT/mammalian target of the rapamycin pathway (PI3K/AKT/mTOR pathway) in GC patients [101]. In particular, according to the molecular analysis performed by the TCGA, *PIK3CA* mutations were reported to be in 42% of the MSI GC [11]. Interestingly, MSI patients with *PIK3CA* mutations were characterised by worse 5-year survival (40%) compared to the MSI group bearing the wild-type gene (70.4%) [102]. *ARID1A* is responsible for chromatin remodelling and together with the negative regulator of the Wnt pathway *RNF43* are often mutated in MSI GC (83% and 55%, respectively) [103,104]. Moreover, the presence of somatic mutations (22%) or loss of expression (35–54%) of *AGO2* and *TNRC6A* genes involved in the microRNA processing machinery have been reported in MSI GC [105].

## 6. EBV-Related Gastric Cancer

Epstein–Barr virus (EBV) emerged as an important virulence factor for nasopharyngeal carcinoma. EBV infection also causes the development of T-cell lymphoma and EBV-associated GC (EBVaGC) [106,107]. Immunotherapy drug treatments were successful against EBV-positive and MSI GCs [108]. Once EBV infects the human body, this does not immediately produce GC and EBV-positive GC is not characterized by any evident clinical manifestations [109,110]. Two theories have been reported about the mechanism of EBV infection. According to the first theory, EBV produces the infection of B-lymphocytes and oral epithelial cells [111]. In particular, since EBV is present in the saliva, this causes the infection of the epithelial cells [111]. The second theory reports EBV reactivation in B-lymphocytes in the stomach and its subsequent release to cause the infection of epithelial cells [111]. EBV infection of lymphocytes results in the interaction of these cells with epithelial cells [112]. This interaction is mediated by integrin β-1/β-2 and the translocation of intracellular adhesion molecule-1 to the cell surface (ICAM-1) produce cell-to cell contact [112]. The virus particles are then internalised by recipient cells through clathrin-mediated endocytosis [112]. EBV-particles inside the host cell nucleus are characterized by a naked, linear DNA genomes and a specific protein capsid protect them [113]. The exposed DNA linear genome is then circularised into a functional chromosome [113]. After circularisation, the chromatinised viral DNA protects it from DNA damage and provides accurate regulation of gene expression [113] (Figure 2). EBV genome is characterised by widely methylated CpG motifs enabling it to establish latent infection [114]. The two types of infection caused by EBV are the lytic and the latent form [114]. The latent infection is the one preferred by EBV and during the long incubation period, EBV causes the methylation of the host DNA and the expansion of GC [113,114]. Latent EBV proteins such as EBERs, BARF-0, EBNA-1, and LMP2A downregulate the miR-200 family causing a reduction in E-cadherin expression [115]. This mechanism is mediated by the upregulation of the E-cadherin repressors, ZEB1 and ZEB2 [115]. Tumour progression involves the loss of cell-to-cell adhesion and this event is also an important step in the carcinogenesis of EBVaGC [115]. EBV is the first human virus expressing many microRNAs [116]. The EBV miRNA BART11 has been shown to downregulate forkhead box protein P1 (FOXP1) transcription factor [117]. FOXP1 downregulation activates the epithelial-mesenchymal transition involving the gastric tumour cells or affecting the tumour microenvironment [117]. This, in turn, accelerates cancer invasion and metastasis, thereby affecting the survival and prognosis of patients [117]. Dong et al. reported that the targeting of APC and Dkk1 by the microRNAs BART10-3p and BART22 leads to activation of the Wnt signalling pathway. Since this specific pathway has a fundamental role in promoting EBVaGC metastasis, these findings are important because provide new prognostic biomarkers and potential therapeutic targets in EBVaGC [118]. Another important component of EBV-positive GC is DNA methylation with 216 genes reported to be downregulated after EBV-infection [119,120]. In particular, *Rec8* methylation levels were higher in EBV-positive than EBV-negative GC tissues and the overall methylation of tissues with no EBV was significantly lower than these subtypes [121]. The anti-tumour effects of Rec8 are caused by downregulation of cell growth-related genes (*G6PD*, *SLC2A1*, *NOL3*, *MCM2*, *SNAI1*, and *SNAI2*) and also by upregulation of both apoptosis or migration inhibitors (*Gadd45G* and *LDHA*) and tumour inhibitors (*PinX1*, *IGFBP3*, and *ETS2*) [121]. Methylation of the *Rec8* gene promoter reduces the survival in patients with GC [121]. Moreover, methylation of different gene promoters can be caused by tf specific proteins expressed during the EBV-incubation period [122]. In particular, one study reported that the activation of *DNMT1* by LMP2A-phosphorylation of STAT3 produces the CpG island methylation of *PTEN* promoter and a consequent loss of PTEN expression in EBV-related GC [122].

## 7. Potential Role and Relevance of POLE/D Mutations in Gastric Cancer

The immune checkpoint inhibitor (ICI) biomarkers approved by the US Food and Drug Administration (FDA) in the treatment of certain cancers are the use of PD-L1 expression, microsatellite instability (MSI)-H/deficient mismatch repair (dMMR) and tumoural mutation burden (TMB) [123,124,125]. Since some responses seen with ICIs do not fully correlate with any of the biomarkers above, other potential predictive biomarkers of response to ICI can exist in GC [126,127]. *POLE* and *POLD1* genes encode for the DNA polymerases ε and δ [127,128]. TMB and clinical benefits of immunotherapy have been associated with mutations in *POLE* and *POLD1* in different cancer types [129,130]. More specifically, Zhu et al. analysed the impact of *POLE* and *POLD1* mutations in gastric adenocarcinoma patients on their prognosis with a potentially beneficial role regarding progression-free and overall survival [131]. The frequency of *POLE*/*POLD1* mutations in GC patients was approximately 8% and the tumours were characterized by adaptive immune resistance tumour microenvironment (TME), deficient mismatch repair (dMMR) status but also by higher PD-L1 expression level, higher TMB, higher MSI and lower aneuploidy score [131]. All of these mechanisms may have important implications for higher responses with ICIs [131]. However, currently the role of *POLE* and *POLD1* mutations in GC as a predictive biomarker of response to immunotherapy remains under research, with no definitive conclusions produced.

## 8. HER2 Overexpression in Gastric Cancer

Human epidermal growth factor receptor 2 (HER2) is a receptor tyrosine kinase proto-oncogene that is increasingly understood to be overexpressed in GC. Different studies have found HER2 overexpression to occur in between 4.4% and 53.4% of GCs, with an average of 17.9% [132]. Prognostically, HER2 appears to be correlated with poorer survival and increased recurrence in GC [133,134]. HER2 targeting through the monoclonal antibody trastuzumab together with chemotherapy has in fact been shown to increase survival in HER2+ GC patients; however, the effect is small with a clinical trial only finding survival increasing from 11.1 months to 13.8 months [135].

During treatment, the tumour microenvironment contributes to tumourigenesis and proliferation but is also associated with immune cell infiltration and treatment efficacy [136]. Suh et al. reported that the HER2 pathway modulates the tumour microenvironment and this is correlated with tumour pathological characteristics and patient survival [137].

HER2 overexpression correlates with PD-L1 overexpression [138]. One study found that *HER2* knockdown in PD-L1/HER2+ GC organoids resulted in a decrease in PD-L1 expression [139]. Furthermore, in co-cultures with immune cells, cytotoxic lymphocytes proliferated more and caused an increase in tumour cell death [139]. Another study reported that 85% of HER2-positive gastric adenocarcinoma (GACs) were characterized by a positive PD-L1 expression and PD-L1 expression positively correlated with HER2 overexpression [138]. These data showed that the combination of anti-PD-L1 treatment with anti-HER2 therapy may have a positive effect in AGC patients with HER2 overexpression [138]. Clinical trials investigating the addition of checkpoint inhibitors to anti-HER2 targeted treatments are ongoing and will be analysed in depth in the following section.

## 9. Immunotherapy in the Clinical Management of Gastric Cancer

Over the last few years, immunotherapy has been actively incorporated as part of first and later lines of systemic treatment for advanced gastric cancer. PD-1 inhibitors have been shown to significantly improve efficacy in several large phase III trials when added to platinum-based chemotherapy, which has for many years been the standard of care in the first line setting for metastatic GC (Figure 3). In this section, we aim to present clinical research data that support a potentially prognostic, as well as predictive role for PD-L1 expression in gastric cancer, and finally present up to date results from large scale, phase III clinical trials, that have investigated the efficacy of PD-1/PD-L1 inhibitors in clinical practice. The description of toxicity of PD-1/PD-L1 inhibitors in gastric cancer trials is not within the scope of this work, as this has been extensively described elsewhere, and does not seem to differ to toxicity observed from the same agents in other tumour types.

### 9.1. PD-L1 as a Prognostic Biomarker

There have been several studies investigating a potential prognostic role for PD-L1 expression in GC, with diverse and controversial results. In a retrospective study by Morihiro, et al., including 283 patients with GC, PD-L1 expression was significantly correlated with a poor prognosis (*p*  =  0.0025). Multivariate analysis revealed that PD-L1 expression was found to be an independent poor prognostic factor, along with diffuse histological type and lymph node involvement. [140]. Furthermore, Chang et al., using tissue microarrays in 464 GC samples, showed that PD-L1 and PD-1 expression was significantly correlated with several adverse prognostic pathologic features, such as T stage, lymphatic invasion and diffuse Lauren histologic type. In the same study, subgroup analyses in which patients were divided into two groups according to CD8+ TILs expression levels (high and low), it was shown that high PD-L1 expression was a negative prognostic factor only in the high CD8+ TILs subgroup [141].

On the contrary, in a study by Cho et al., including 78 MSI-H GC tissue samples, immune cell PD-L1 expression was often associated with intestinal histologic type, with a lower risk of lymph node involvement and lower tumour stages, as compared to MSI-H GCs with negative PD-L1 expression. Furthermore, immune cell PD-L1 expression was independently a favourable prognostic factor for overall survival (OS) (versus PD-L1 negative; hazard ratio, 3.451; 95% confidence interval, 1.172–12.745; *p* = 0.025) [142]. Similarly, in a study by Liu et al., including 598 surgically resected primary GC samples, PD-L1 expression was identified as a positive prognostic factor in this large cohort of patients [143].

Little is known about the prognostic role of PD-L1 expression in the rare subset of EBVaGC, which is known to be highly immunogenic and is associated with a high response rate to immunotherapy, to the same extent as MSI-H GCs. A small study focusing on EBV-positive GCs failed to identify any significant prognostic roles for either PD-L1 or PD-L2 expression [144]. However, in a more recent study in EBVaGC cases, PD-L1 positivity was associated with favourable clinicopathological features, and was independently a predictor for longer disease-free survival (hazard ratio [HR] and 95% CI: 0.45 [0.22–0.92], *p* = 0.03) and overall survival (HR and 95% CI: 0.17 [0.06–0.43], *p* < 0.001) [145].

As the controversy around any potential prognostic role for PD-L1 expression is becoming obvious, a systematic review and meta-analysis by Gu et al., included a total of 15 eligible studies of 3291 patients, in an effort to provide more collective evidence. Eleven studies concluded that PD-L1 overexpression was associated with poor prognosis of GC, whereas three studies revealed that PD-L1 overexpression was associated with improved prognosis, and there was no association in one study. On aggregate, this meta-analysis showed that PD-L1 expression was a significant negative prognostic factor for overall survival (HR = 1.46, 95%CI = 1.08–1.98, *p* = 0.01, random-effect). Furthermore, higher PD-L1 expression was associated with the depth of invasion, venous invasion, EBV infection, lymph-node involvement, and MSI-status [146].

The controversial results of all relevant studies investigating PD-L1 expression as a prognostic factor are likely subject to several study limitations, including the small number of participants, different cut-off levels of PD-L1 expression included, different quantification methods measuring expression on tumour cells and/or infiltrating immune cells, different monoclonal/polyclonal PD-L1 antibodies, as well as all biases related to the retrospective design of the studies above. Therefore, at present, we can say that there is no robust or definitive evidence to support any substantial prognostic role for PD-L1 expression in GC.

A more convincing, although not conclusive, predictive role for PD-L1 expression, as measured by the combined positive score (CPS), has become increasingly evident in clinical trials investigating immunotherapy in the first or subsequent lines of treatment in GC, as an increasing CPS has been associated with improved outcomes with PD-1 inhibitors. In the following section, we aim to describe the practice-changing phase III trials that have established immunotherapy in the management of advanced gastric/gastro-oesophageal junction (GOJ) cancer (Table 1), where most of the evidence for a predictive role for PD-L1 CPS is derived by post hoc analyses. Of note, although GOJ cancers were also included in trials investigating the efficacy of immunotherapy in oesophageal cancer, these trials were excluded from our descriptive analysis due to the fact that the GOJ cancers included in these trials were mainly Siewert type 1, which are largely managed as distal oesophageal cancers, and are also believed to behave biologically similar to oesophageal carcinomas, as opposed to GC.

### 9.2. Immunotherapy in Chemo-Resistant/Refractory Setting; PD-L1 Expression as a Predictive Biomarker

One of the first phase III trials that investigated the use of PD-1 monotherapy in the chemo-refractory setting in metastatic GC was the ATTRACTION-2 trial, which randomised previously treated patients with metastatic gastric or gastro–oesophageal junction (GOJ) adenocarcinoma with more than two lines of treatment, to nivolumab or placebo. Median overall survival was 5.26 months (95% CI 4.60–6.37) in the nivolumab group and 4.14 months (3.42–4.86) in the placebo group (hazard ratio 0.63, 95% CI 0.51–0.78; *p* < 0.0001). [147]. A more recent update of the trial with 2-year follow-up data confirmed the long-term benefit of nivolumab over placebo, with a higher OS rate noted in the nivolumab vs. placebo group at 2 years (10.6% vs. 3.2%). The OS benefit was observed regardless of tumour PD-L1 expression, although this must be interpreted with caution because only 192 of 493 patients (39%) had PD-L1 results available for analysis [148]. Despite the fact that the main population of the study was Asian patients, the positive benefit of nivolumab in overall survival established this option as a standard third line and beyond option regardless of PD-L1 expression levels in many countries.

Keynote-061 was a phase III trial of pembrolizumab which investigated the efficacy of PD-1 monotherapy as a second line option and comparing it to standard of care taxanes. The study randomised patients with advanced gastric or GOJ adenocarcinoma, who previously failed 5-FU/platinum-based chemotherapy, to receive either pembrolizumab or paclitaxel. Primary outcomes were overall survival (OS) and progression-free survival (PFS) in patients with a PD-L1 CPS of 1 or higher. Median overall survival was 9.1 months with pembrolizumab and 8.3 months with taxol (hazard ratio [HR] 0.82, 95% CI 0.66–1.03; one-sided *p* = 0.0421). Median progression-free survival was 1.5 months with pembrolizumab and 4.1 months with taxol (HR 1.27, 95% CI 1.03–1.57). Despite the fact that this trial failed its primary outcomes, a post hoc unplanned analysis focusing on the subgroup of patients with a PD-L1 CPS ≥ 10, demonstrated a numerical survival benefit for pembrolizumab over taxol with a hazard ratio (HR) of 0.64 (95% CI 0.41–1.02) and a median overall survival of 10.4 vs. 8 months [149]. A more recent update of the trial with 2-year follow-up data showed that pembrolizumab increased the OS benefit amongst patients with higher PD-L1 expression (CPS ≥ 5: HR, 0.72, 24-month rate, 24.2% vs. 8.8%; CPS ≥ 10: 0.69, 24-month rate, 32.1% vs. 10.9%) [150]. Finally, although not formally reported, inspection of the survival curves for the subgroup of patients with CPS < 1 in KEYNOTE-061, suggests a detriment for patients treated with pembrolizumab compared to paclitaxel. In conclusion, despite the fact that Keynote-061 was in fact a negative trial, it provided valuable evidence for an improved efficacy with immunotherapy in a PD-L1 enriched subgroup of patients with advanced GC, which formed the basis for a more appropriate design of subsequent large phase III trials of immunotherapy in this disease.

A similar design phase III trial, the JAVELIN Gastric 300, randomised patients who failed two prior lines of treatment to receive either avelumab or physician’s choice of chemotherapy (paclitaxel or irinotecan). The primary outcome was overall survival. The trial failed its primary outcome of improving OS [median, 4.6 versus 5.0 months; hazard ratio (HR) = 1.1 [95% confidence interval (CI) 0.9–1.4]; *p* = 0.81] and the secondary outcome of PFS [median, 1.4 versus 2.7 months; HR = 1.73 (95% CI 1.4–2.2); *p* > 0.99] or ORR (2.2% vs. 4.3%) in the avelumab versus chemotherapy arms, respectively [151]. In this trial, PD-L1 expression was defined as positive or negative on the basis of the expression of at least ≥1% on tumour cells. A total of 23% of samples were considered PD-L1-positive, while there was no difference between PD-L1 positive and negative subgroups for either OS (4.0 vs. 4.6 months) or PFS (1.4 vs. 1.4 months) in the Avelumab arm [127]. This trial failed to demonstrate any signals of efficacy of avelumab in the chemo-refractory setting, regardless of PD-L1 expression levels, which could be attributed to the use of TPS as opposed to CPS in the sub-group analysis, and also possible inherent biological differences between avelumab (PD-L1) vs. other PD-1 inhibitors.

### 9.3. Immunotherapy in Previously Untreated Patients; PD-L1 Expression as a Predictive Biomarker

The next logical step was to investigate the incorporation of immunotherapy in the first line setting, in previously untreated patients, with advanced gastric cancer. One of the first phase III trials that attempted to answer this specific question was the Keynote-062 trial. In this trial, patients were randomized 1:1:1 to pembrolizumab, pembrolizumab plus chemotherapy (cisplatin plus fluorouracil or capecitabine) or chemotherapy plus placebo. Primary end points were OS and PFS in patients with PD-L1 CPS ≥ 1 or ≥10. This trial demonstrated non-inferiority for pembrolizumab to chemotherapy for OS in patients with CPS ≥ 1 (median, 10.6 vs. 11.1 months; hazard ratio [HR], 0.91; 99.2% CI, 0.69–1.18), however pembrolizumab monotherapy was not found to be superior to chemotherapy in patients with CPS ≥ 1. In patients with CPS ≥ 10, pembrolizumab numerically prolonged OS vs. chemotherapy (median, 17.4 vs. 10.8 months; HR, 0.69; 95% CI, 0.49–0.97); however, this difference was not tested for statistical significance. Pembrolizumab combination with chemotherapy was not found to be superior to chemotherapy for OS in patients with CPS ≥ 1 (12.5 vs. 11.1 months; HR, 0.85; 95% CI, 0.70–1.03; *p* = 0.05) or CPS ≥ 10 (12.3 vs. 10.8 months; HR, 0.85; 95% CI, 0.62–1.17; *p* = 0.16) or for PFS in patients with CPS ≥ 1 (6.9 vs. 6.4 months; HR, 0.84; 95% CI, 0.70–1.02; *p* = 0.04) [152]. In a more recent update of the trial in ASCO GI 2022, with a median follow-up of 54.3 months (range, 46.8–66.1), efficacy was consistent with the initial final analysis data; 24-month OS rates (pembrolizumab vs. chemotherapy) were 26.6% versus 18.8% in the CPS ≥ 1 population and 39.1% versus 21.1% in the CPS ≥ 10 population. Twenty-four-month OS rates (pembrolizumab + chemotherapy vs. chemotherapy) were 24.5% versus 18.8% in the CPS ≥ 1 population and 28.3% versus 21.1% in the CPS ≥ 10 population [153]. Although Keynote-062 was considered a negative trial, it provided further valuable supporting evidence with regards to the potential predictive role of PD-L1 CPS in terms of the selection of patients that would most benefit from immunotherapy, namely patients with PD-L1 CPS ≥ 1, and ideally PD-L1 CPS ≥ 10.

A role for immunotherapy as maintenance strategy after first line platinum-based chemotherapy has also been investigated in the phase III JAVELIN Gastric 100 trial, where patients without progressive disease after induction chemotherapy of 12 weeks duration with oxaliplatin plus a fluoropyrimidine were randomised 1:1 to avelumab or continued chemotherapy. The primary outcome was OS in all patients or the PD-L1-positive subgroup (≥1% of tumour cells). Median OS was 10.4 months (95% CI, 9.1 to 12.0 months) versus 10.9 months (95% CI, 9.6 to 12.4 months); in the PD-L1-positive subgroup (*n* = 54), the HR for OS was 1.13 (95% CI, 0.57 to 2.23; *p* = 0.6352). In an exploratory analysis amongst the PD-L1-positive patients, using PD-L1 CPS ≥ 1 as a cut-off (*n* = 137), median OS was 14.9 months (95% CI, 8.7 to 17.3 months) with avelumab versus 11.6 months (95% CI, 8.4 to 12.6 months) with chemotherapy (HR, 0.72; 95% CI, 0.49 to 1.05) [154]. This trial failed to demonstrate a role for immunotherapy (PD-L1 inhibitor) as switch maintenance treatment in the first line setting; however, a numerical favourable survival benefit was seen when CPS was used to quantify PD-L1 expression, for PD-L1 positive tumours, thus contributing to the already existing literature highlighting PD-L1 CPS as a more accurate predictive biomarker for immunotherapy efficacy in gastric cancer.

The first phase III trial that demonstrated a clear survival benefit for the addition of PD-1 inhibitors in the first line platinum-based treatment of gastric cancer was the Checkmate-649. Treatment naive, unresectable, HER2 negative gastric, gastro-oesophageal junction, or oesophageal adenocarcinoma, regardless of PD-L1 expression, were randomly assigned (1:1:1) to nivolumab plus chemotherapy (capecitabine and oxaliplatin or leucovorin, fluorouracil, and oxaliplatin), nivolumab plus ipilimumab, or chemotherapy alone. Primary endpoints were OS and PFS for nivolumab plus chemotherapy versus chemotherapy alone in patients with a PD-L1 CPS ≥ 5. Nivolumab plus chemotherapy was associated with a significant improvements in OS (hazard ratio [HR] 0.71 [98.4% CI 0.59–0.86]; *p* < 0.0001), with a median OS of 14.4 vs. 11.1 months, and PFS (HR 0.68 [98% CI 0.56–0.81]; *p* < 0.0001), versus chemotherapy alone in patients with a PD-L1 CPS ≥ 5. Further analysis demonstrated significant improvement in OS (median OS of 14 vs. 11.3 months, HR: 0.77, 99% CI 0.64–0.92, *p* = 0·0001), and PFS in patients with a PD-L1 CPS ≥ 1, and all randomly assigned patients (median OS of 13.8 vs. 11.6 months, HR: 0.8, 99% CI 0.68–0.94, *p* = 0.0002) [155]. In a more recent update of the trial from ASCO GI 2022, with 2-year follow up data, there was a continued maintained benefit in OS, whereas the HR (95% CI) for OS was 0.66 (0.56–0.77) among pts with a PD-L1 CPS ≥ 10 (median OS for nivolumab + chemotherapy vs. chemotherapy: 15.0 [95% CI 13.7–16.7] vs. 10.9 [95% CI 9.8–11.9] months) [156]. Most recently, the analysis of the nivolumab-ipilimumab vs. chemotherapy arms came to light in which nivolumab plus ipilimumab failed to improve overall survival in patients with PD-L1 CPS ≥ 5 versus chemotherapy alone (median overall survival was 11.2 (95% confidence interval 9.2, 13.4) versus 11.6 (95% confidence interval 10.1, 12.7) months, respectively (hazard ratio 0.89; 96.5% confidence interval 0.71, 1.10; *p* = 0.2302) [157]. Overall, these results have established a new standard of care in the first line setting for advanced gastric cancer, highlighting the significance of PD-L1 CPS as a selection biomarker for patients that may benefit from the addition of a PD-1 inhibitor to chemotherapy.

A similar design trial to the Checkmate-649, the ATTRACTION-4, randomized patients to chemotherapy (oxaliplatin plus either oral S-1 [SOX] or oral capecitabine [CAPOX]), in addition to either nivolumab or placebo. The primary endpoints were PFS and OS in the intention-to-treat population, which included all randomised patients, without patient preselection based on PD-L1 expression. Median PFS was 10.45 months (95% CI 8.44–14.75) in the nivolumab plus chemotherapy group and 8.34 months (6.97–9.40) in the placebo plus chemotherapy group (hazard ratio [HR] 0.68; 98.51% CI 0.51–0.90; *p* = 0.0007), which was statistically significant. Median OS was 17.45 months (95% CI 15.67–20.83) in the nivolumab plus chemotherapy group and 17.15 months (15.18–19.65) in the placebo plus chemotherapy group (HR 0.90; 95% CI 0.75–1.08; *p* = 0.26) [158], which failed statistical significance.

Similar positive results were obtained from the initial analysis of the ORIENT-16 trial, conducted in China. This trial was a randomized, double-blind, phase III trial investigating the efficacy of sintilimab (PD-1 inhibitor) combined with chemotherapy vs. chemotherapy for treatment naïve patients with metastatic gastric/GOJ adenocarcinoma. Patients were randomized 1:1 to receive sintilimab or placebo plus chemotherapy (oxaliplatin and capecitabine). The primary endpoints were OS in patients with CPS ≥ 5 and all randomized patients. Chemotherapy in combination with sintilimab was associated with superior OS in both CPS ≥ 5 (median 18.4 vs. 12.9 months; HR 0.660; *p* ≥ 0.0023) and all randomized patients (median 15.2 vs. 12.3 months; HR 0.766; *p* ≥ 0.0090), with longer PFS and a higher ORR in favour of the interventional arm with sintilimab [159].

Both the ATTRACTION-4 and ORIENT-16 trial results overall support the efficacy of the addition of immunotherapy to first line platinum-based chemotherapy in advanced gastric cancer patients as a new standard of care, with most of the benefit seen in PD-L1 CPS positive disease.

### 9.4. Immunotherapy in HER2-Positive Gastric Cancer

PD-1 inhibition is thought to work synergistically with HER2 inhibition increasing ADCC (antibody-dependent cellular cytotoxicity), and this concept has been investigated in the Keynote-811 trial, where treatment naive patients with HER2-positive GC or GOJ adenocarcinoma were randomized to receive pembrolizumab or placebo plus trastuzumab and investigator’s choice of fluorouracil/cisplatin or capecitabine/oxaliplatin. Overall, 84% of patients had a PD-L1 CPS ≥ 1. Confirmed ORR (95% CI) was 74.4% (66.2–81.6) for pembrolizumab + chemotherapy vs. 51.9% (43.0–60.7) for placebo + chemotherapy, *p* = 0.00006); CR (complete response) rate was 11.3% vs. 3.1% and DCR (disease control rate) (95% CI) was 96.2% (91.4–98.8) vs. 89.3 (82.7–94.0). [160]. The very encouraging preliminary results from the Keynote-811 provide a proof of concept for the synergistic effect of PD-1 and HER2 inhibition in HER2-positive gastric cancer, which stands a very good chance to consist of a new standard of care in the first line setting, providing long-term analysis also reveals a survival benefit for pembrolizumab, trastuzumab and chemotherapy in this setting.

### 9.5. Immunotherapy Efficacy in Rare Subgroups: Role of MSI, EBV and TMB (Tumour Mutational Burden) Status as Predictive Biomarkers

The role of immunotherapy in the management of mismatch-repair deficient (MMR-D) and EBV-associated gastric tumours has long been established through several studies. In stage IV advanced gastric cancer, EBV-positive and MMR-D tumours are identified both in approximately 6% of cases [161]. In the Keynote-158 phase II trial investigating the efficacy of pembrolizumab in MMR-D tumours, 11 of 24 patients with advanced gastric cancer demonstrated an objective response as their best response (ORR, 45.8%), which was associated with a median PFS of 11.0 months and a median OS which was not reached [162]. In a post hoc analysis of the outcomes of MSI-H GC patients included in the phase 2 single-arm trial KEYNOTE-059 (≥3 lines of treatment) and the phase 3 KEYNOTE-061 (second line) and KEYNOTE-062 (first line treatment) trials, the median overall survival was not reached (NR) for pembrolizumab in all three of these trials. The median PFS) for pembrolizumab was NR (95% CI, 1.1 months to NR) in KEYNOTE-059 and 17.8 months (95% CI, 2.7 months to NR) in KEYNOTE-061 (vs. 3.5 months for chemotherapy). In KEYNOTE-062, the median PFS was 11.2 months (95% CI, 1.5 months to NR) for pembrolizumab, NR (95% CI, 3.6 months to NR) for pembrolizumab plus chemotherapy, and only 6.6 months (95% CI, 4.4–8.3 months) for chemotherapy alone. The ORR for pembrolizumab was 57.1% in KEYNOTE-059 and 46.7% (vs 16.7% for chemotherapy alone) in KEYNOTE-061. In KEYNOTE-062, the ORR was 57.1% for pembrolizumab, 64.7% for pembrolizumab plus chemotherapy, and 36.8% for chemotherapy alone [163]. Very few clinical trial data exist in regards to the efficacy of immunotherapy in EBV-positive GC; in a phase II study of pembrolizumab in metastatic GC patients with MSI-H and Epstein–Barr virus-positive tumours, impressive responses to pembrolizumab were noted (overall response rate (ORR) 85.7% in MSI-H metastatic GC and ORR 100% in Epstein–Barr virus-positive tumours) [164]. The results above unequivocally establish MSI-H status and EBV positivity as by far the most robust biomarkers of response to immunotherapy in GC, however it has to be noted that despite the high and durable responses, there is a significant fraction of patients which test positive for the above biomarkers that do not respond to immunotherapy, and therefore there is a great unmet need for further molecular subclassification of these gastric cancer subtypes in order to elucidate mechanisms of primary resistance to immunotherapy.

In GC, ~8% of tumours are associated with high TMB, defined as >17 mut/MB. Of the 8% of GCs that were high TMB, the majority of them was due to microsatellite instability. Amongst microsatellite stable (MSS) tumours, only 1.7% of GC was found to have high TMB [126]. Clinical data regarding the efficacy of immunotherapy in GC with high TMB are derived as exploratory analyses of few randomised trials. In the Keynote-061 trial, Foundation One CDx-measured TMB (with a cut off of 10 mut/MB) showed a positive correlation between high TMB tumours and ORR, PFS, and OS in patients treated with pembrolizumab, but not PFS or OS in patients treated with paclitaxel [165]. In the phase III KEYNOTE-062 study, the overall prevalence of TMB of 10 mut/Mb or more was 16% between treatment groups; 44% of patients who had TMB ≥ 10 mut/Mb had MSI-H tumours. An improvement in ORR, PFS, and OS was noted in patients with high TMB (≥10 mut/Mb) treated with pembrolizumab. However, when the analysis excluded the MSI-H subgroup of patients, the positive correlation between clinical outcomes with pembrolizumab or pembrolizumab plus chemotherapy and TMB as a continuous variable versus chemotherapy alone was attenuated [166]. All the data above collectively indicate a potential, but currently not adequately convincing role for TMB as a predictive biomarker for immunotherapy response in GC, warranting further investigation within prospective randomised immunotherapy trials.

It is becoming obvious that there are still several unanswered questions and unmet needs in regards to optimising immunotherapy efficacy in GC, and therefore there is ongoing active research with several immunotherapy agents being tested in various indications, i.e., a second line phase III trial of QL1604 (anti-PD-1 antibody) plus nab-paclitaxel versus paclitaxel (NCT04435652), phase III trials of pembrolizumab or durvalumab along with chemotherapy in the peri-operative setting in GC (NCT03221426, NCT04592913), as well as a phase III trial of the addition of nivolumab to adjuvant chemotherapy in resected GC (NCT03006705). In the HER2-positive setting, trastuzumab deruxtecan is being studied in several combinations, amongst which is pembrolizumab, in the DESTINY-Gastric03 study. Combinations of immunotherapy with multi-tyrosine kinase inhibitors (TKI) are also under investigation in several trials, i.e., a phase III trial of regorafenib plus nivolumab vs. chemotherapy in third- or later-line (NCT04879368), and a phase III trial of lenvatinib plus pembrolizumab plus chemotherapy as induction, followed by lenvatinib plus pembrolizumab as maintenance versus chemotherapy alone (NCT04662710). Finally, chimeric antigen receptor T (CAR-T) cell therapies targeting claudin18.2 (CLDN18.2) are also under early clinical development in advanced gastric cancer, after initial promising efficacy seen in a phase 1 trial [167].

## 10. Conclusions

In conclusion, immunotherapy has been incorporated in the clinical management of advanced GC based on recent positive phase III clinical trial results. Most of the benefit is seen in PD-L1 CPS positive, MSI-H/MMR-D, EBV-positive, and possibly TMB-high tumours. Although these biomarkers have been associated with durable responses to immunotherapy, there is still a considerable proportion of patients who fail to respond, while ultimately cancer cells develop immune-evasion strategies and develop resistance. Further clinical combined with translational research is eagerly needed to optimise immunotherapy efficacy and overcome emerging PD-1/PD-L1 resistance in order to improve GC patients’ outcomes.

## Figures and Tables

**Figure 1 cancers-14-04378-f001:**
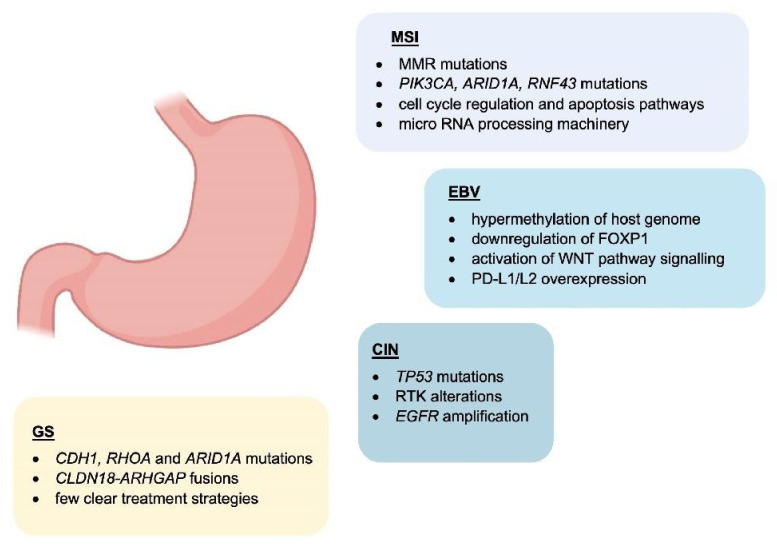
Main features of GC subtypes. Schematic representation of the molecular characteristics associated with GC molecular subtypes.

**Figure 2 cancers-14-04378-f002:**
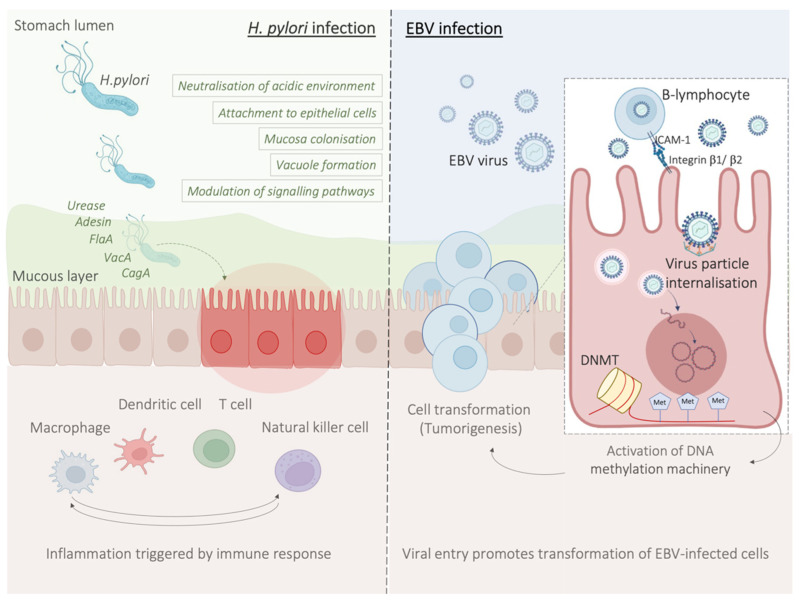
*H. pylori* and EBV mechanisms of infection at a glance. *H. pylori* infection causes a local inflammation state with consequent infiltration of inflammatory cells, and increased risk of gastric carcinogenesis (**left panel**), EBV infection process, associated with development of EBV-associated GC (**right panel**).

**Figure 3 cancers-14-04378-f003:**
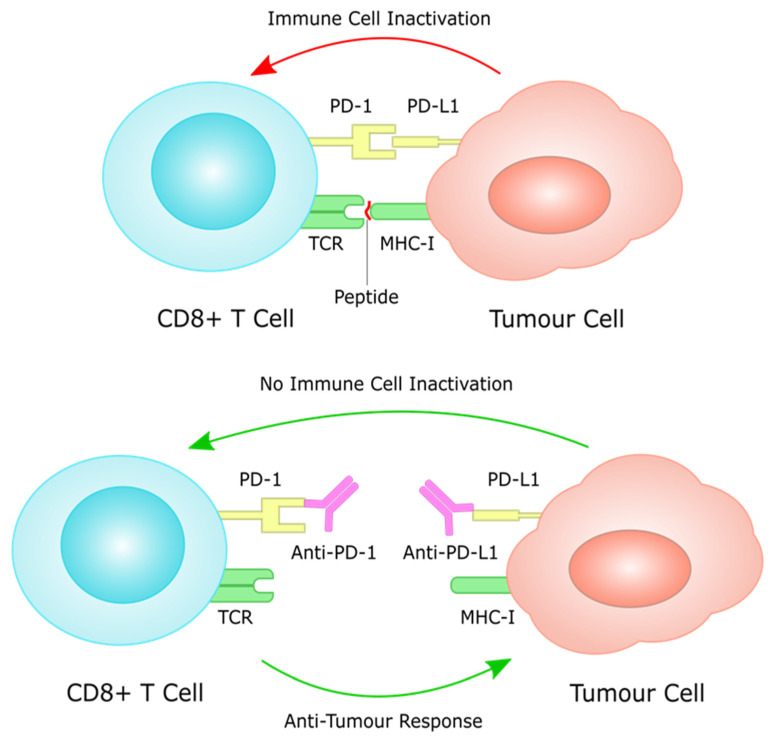
PD-L1 on tumour cells works through the PD-1 receptor on T cells to induce immune cell inactivation. Treatment with immune checkpoint inhibitors inhibits this interaction, often through monoclonal antibodies against PD-1 and PD-L1, restoring immune cell function against tumour cells.

**Table 1 cancers-14-04378-t001:** Summary of phase 3 trials of immunotherapy in gastric cancer.

Trial	Setting	Interventional Arm	Comparator	Outcomes	Comments
ATTRACTION-2	Chemo-refractory	nivoluab	placebo	Positive for OS	-
Keynote-061	2nd line	pembrolizumab	taxol	No difference for OS	Potential benefit in CPS ≥ 10
JAVELIN Gastric 300	3rd line	avelumab	Taxanes/irinotecan	No difference for OS	-
Keynote-062	1st line	pembro or pembro+ chemo	Platinum chemo +placebo	Pembro non inferior to chemo in CPS ≥ 1	Pembro numerically superior to chemo in CPS ≥ 10
JAVELIN Gastric 100	1st line maintenance	avelumab after chemo induction	Chemotherapy continuation	No difference for OS	Avelumab numerically superior in CPS (+)
Checkmate-649	1st line (CPS ≥ 5)	nivo+ chemo or nivo-ipi	Platinum chemo	Nivo+chemo superior OS in CPS ≥ 5	Nivo-ipi: no difference for OS
ATTRACTION-4	1st line (Asian population)	Nivo + chemo	Platinum chemo + placebo	Nivo+ chemo superior for PFS regardless of PD-L1 expression	No difference in OS
ORIENT-16	1st line in CPS ≥ 5 (Asian population)	Sintilimab + chemo	Platinum chemo + placebo	Sintilimab + chemo superior for OS in CPS ≥ 5 and all randomized patients	-
Keynote-811	1st line HER2 positive	Pembro+trastuzumab + chemo	placebo+trastuzumab + chemo	Pembro arm superior for ORR	Very preliminary data

## Data Availability

No new data were created or analysed in this study. Data sharing is not applicable to this article.

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
