# Peer review of "Incorporating Immunotherapy in the Management of Gastric Cancer: Molecular and Clinical Implications"

_cancers, 2022, doi:10.3390/cancers14184378_

Round 1

Reviewer 1 Report

This is a well written paper by Agnarell et al discussing the   molecular and clinical implications of  immunotherapy in the management of gastric cancer: 

There are no major comments for this paper; it is very straightforward and it summarizes almost everything related to the topic very clearly.

Minor comments:

-The abstract and introduction prepare the reader for a review about immunotherapies in GC; however, significant part of the paper tackle GC etiology. I suggest rephrasing the aims of this review

-Figure 2 is not coherent. It shows mode of entry for EBV, whereas for H. pylori, it shows the mode of action. I suggest illustrating both mode of entry and mode of action for EBV and H. pylori

-Figure 3 is confusing and the “peptide” is not explained in the legend. I suggest dividing the figure into two panels; one showing the interaction between tumor cell and T-cell in normal state, and the other panel showing these cells in the presence of PD-1 and PD-L1 inhibitors.

-The paper discusses thoroughly the effects of immunotherapy on GC in a biased way as the authors did not talk about the side-effects of such approach. Along with targeting the cancer, immunocomprimization can be very problematic. It is worth addressing the limitations and draw-backs of immunotherapy as well.    

-The English in the abstract and introduction needs revision for sentence structure

Author Response

Reviewer #1
We appreciate the reviewer’s comments that: ‘This is a well written paper’ and that ‘...it is very straightforward and it summarizes almost everything related to the topic very clearly’. Following the reviewer’s suggestions, as explained below, we have clarified the issues that were raised in our revised version. More precisely:

1) The abstract and introduction prepare the reader for a review about immunotherapies in GC; however, significant part of the paper tackle GC etiology. I suggest rephrasing the aims of this review.
As suggested by the Reviewer, we have now rephrased the aims of this review by editing the abstract and the introduction accordingly.

2) Figure 2 is not coherent. It shows mode of entry for EBV, whereas for H. pylori, it shows the mode of action. I suggest illustrating both mode of entry and mode of action for EBV and H. pylori.
We would like to thank the reviewer for this constructive suggestion. As requested, we have now implemented the changes.
We agree that including both mode of entry and mode of action for H. pylori and EBV may indeed provide a clearer overview. (Line 277).

3) Figure 3 is confusing and the “peptide” is not explained in the legend. I suggest dividing the figure into two panels; one showing the interaction between tumor cell and T-cell in normal state, and the other panel showing these cells in the presence of PD-1 and PD-L1 inhibitors.
Thank you for pointing this out. We have now edited Figure 3 accordingly, by dividing it into two panels. (Line 503).

4) The paper discusses thoroughly the effects of immunotherapy on GC in a biased way as the authors did not talk about the side-effects of such approach. Along with targeting the cancer, immunocomprimization can be very problematic. It is worth addressing the limitations and draw-backs of immunotherapy as well.
We would like to thank you for this observation and suggestion. To make this aspect clearer we have introduced the following paragraph at the beginning of paragraph 9 (Line 496):
“Over the last few years, immunotherapy has been actively incorporated as part of first and later lines of systemic treatment for advanced gastric cancer. PD-1 inhibitors have been shown to significantly improve efficacy in several large phase III trials when added to platinum-based chemotherapy, which has for many years been the standard of care in the first line setting for metastatic GC (Figure 3). In this section, we aim to present clinical research data that support a potentially prognostic, as well as predictive role for PD-L1 expression in gastric cancer, and finally present up to date results from large scale, phase III clinical
trials, that have investigated the efficacy of PD-1/PD-L1 inhibitors in clinical practice. The description of toxicity of PD-1/PD-L1 inhibitors in gastric cancer trials is not within the scope of this work, as this has been extensively described elsewhere, and does not seem to differ to toxicity observed from the same agents in other tumour types.”).

5) The English in the abstract and introduction needs revision for sentence structure.
As suggested, we have now revised the English in the abstract and introduction. 

see attached file

Reviewer 2 Report

The Authors outlined, within a comprehensive and interesting review, the current knowledge immunotherapy for gastric cancer, highlighting molecular classification, clinical benefits, toxicity profile, limitation in the current indications and combination strategy. The manuscript is of interest. Few comments below.

1)     I feel the manuscript has been organized and drafted mostly as a qualitative review, which is fine. I would, however, try to improve the methodology employed.

2)     Please refer to the PICOTS criteria and frame populations, intervention(s), comparator(s), outcomes, timing, setting.

3)     Please provide details of the search strategy employed in the present review paper.

4)     Please provide details in the search strategy. I would suggest referring to the PRISMA criteria (please add a PRISMA flow-chart).

5)     I would suggest reducing the textual part and add tables to wrap up the main findings and help the reader navigating the data presented.

Author Response

Reviewer #2
We would like to thank the reviewer for the comment that this is: ‘… a comprehensive and interesting review’ and that: ‘The manuscript is of interest’. We appreciate the reviewer’s comments, which we addressed as described below:

1) I feel the manuscript has been organized and drafted mostly as a qualitative review, which is fine. I would, however, try to improve the methodology employed.
Following the Reviewer’s suggestions to improve the methodology employed in this review, Please see the responses below for point 2, 3 and 4.

2) Please refer to the PICOTS criteria and frame populations, intervention(s), comparator(s), outcomes, timing, setting.
3) Please provide details of the search strategy employed in the present review paper,
4) Please provide details in the search strategy. I would suggest referring to the PRISMA criteria (please add a PRISMA flowchart).

Thank you for your suggestions. This manuscript consists of a detailed narrative description of the most significant phase III clinical trials which have investigated a role for modern immunotherapy in the management of gastric cancer patients, in both previously treated, and treatment-naive patients. We feel that the proposed PICOTS and PRISMA criteria are far more relevant
for a manuscript that would aim to conduct a systematic review or meta-analysis, which is not the scope of our current review.

5) I would suggest reducing the textual part and add tables to wrap up the main findings and help the reader navigating the data presented.
Thank you for this comment. We have already included a table summarizing the results of the most significant large scale clinical trials of immunotherapy in gastric cancer. Due to the narrative nature of our manuscript, we feel that there is no effective way to reduce the textual part, and instead include this information in further tables embedded within the manuscript.

see attached file

Round 2

Reviewer 2 Report

None